# Transformative Business Models for Decarbonization: Insights from Prize-Winning Start-Ups at the Web Summit

**Evaldo Costa** [1,*] , **Margarida Fontes** [2] **and Nuno Bento** [1]

1 DINÂMIA'CET-Iscte, Centro de Estudos Sobre a Mudança Socioeconómica e o Território, Iscte-Instituto Universitário de Lisboa, 1649-026 Lisboa, Portugal
2 LNEG—Laboratório Nacional de Energia e Geologia, 1649-038 Lisboa, Portugal
* Correspondence: jose.costa@iscte-iul.pt

**Abstract:** The increasing social pressure for decarbonization has placed businesses under considerable scrutiny to actively reduce carbon emissions. A critical step towards achieving this objective is to shift conventional production and consumption systems to more sustainable alternatives. Thus, there is an emergent need to understand the patterns and drivers of the transformative business models (BMs) that underpin that shift. This study adopts a mixed-methods approach that integrates different literature streams—including Sustainability Transitions Theory (STT), Strategic Niche Management (SNM), and the Business Models approach—and stakeholders' interviews to investigate the key elements of business models that lead towards sustainable practices. This research examines the organizational arrangements of European start-ups operating between 2014 and 2020. The transformation towards decarbonized production and consumption is characterized by an efficient combination of business strategies that incorporate advanced technologies (ATs), such as artificial intelligence (AI), machine learning (ML) and its algorithms, along with sustainable elements, resulting in transformative business models. By exploring the driving elements behind the transition to low-carbon approaches, this study fills a significant gap in the existing literature on business models. The findings from this research also hold relevance for policymakers to promote decarbonization.

**Keywords:** decarbonization; sustainable alternatives; transformative business models (TBM); Sustainability Transitions Theory (STT); Strategic Niche Management (SNM); advanced technologies; start-ups; sustainable practices; carbon emissions; sustainable development goals (SDGs)

## 1. Introduction

The urgent need to decarbonize economic sectors to mitigate climate risks and achieve the goals outlined in the Paris Agreement is becoming increasingly apparent as human activities continue to contribute to rising carbon emissions [1]. Within this context, the business sector plays a critical role in supporting the transition towards low-carbon production and consumption. The integration of advanced technologies (AT), sustainable mechanisms, and transformative business models (TBMs) has gained prominence on the global sustainable development agenda [2].

Efforts to harmonize environmental, economic, and social considerations in business pose complex challenges. In response, innovative organizations, including start-ups, are actively incorporating ATs, exploring new institutional arrangements, and fostering innovation to transform their BMs. These transformations aim to create, deliver, and capture sustainable value while simultaneously striking a balance that enables effective capitalization [3].

The hypothesis of the study is that start-up BMs that integrate advanced technologies and sustainable elements are crucial drivers of transformative changes in production and consumption methods. These changes, achieved through a combination of innovative strategies and institutional arrangements, lead to transformative sustainable business

models (TSBMs), which accelerate the shift towards a low-carbon economy. By analyzing these transformations using frameworks like Sustainability Transitions Theory (STT) and Strategic Niche Management (SNM), the study aims to uncover the key elements and pathways that enable sustainable practices among start-ups.

However, the existing literature on how organizations and their BMs can effectively ease the decarbonization of production and consumption remains incomplete and marked by controversies [4]. Ongoing debates revolve around the essential elements and avenues to be explored to establish new institutional arrangements that drive the transformative sustainable business models (TSBMs), ultimately accelerating the transition to a low-carbon economy.

To address these gaps and investigate the key elements of BMs in start-ups that drive changes in production and service delivery methods relevant to decarbonization, this study adopts different theoretical frameworks that integrate the Sustainability Transitions Theory (STT), Strategic Niche Management (SNM), and the BM approach, combined with stakeholder interviews to explore the elements of TSBMs in the context of start-ups. The research objective regarding the TSBMs in start-ups is twofold: (1) to identify prevailing BMs that are facilitating transformation to accelerate decarbonization, and (2) to determine the key elements that characterize and measure TBMs, supporting transitions towards sustainability.

This article is organized into five sections. The subsequent section offers a review of the relevant streams of the literature for the study of TBM, including STT, SNM, TBM, and ATs such as the Internet of Things (IoT). Artificial Intelligence (AI) and its algorithms, Machine Learning (ML), and sustainable elements within a novel institutional framework. The third section details the methodology employed in this study. The fourth section presents the results, while the fifth section offers the conclusions, including limitations and recommendations for future studies.

## 2. Conceptual Framework

The urgent need to address climate change and the risks associated with carbon emissions has placed significant pressure on businesses to actively contribute to global decarbonization efforts to reduce or eliminate emissions of carbon dioxide ($CO_2$) and other greenhouse gas (GHG) emissions in production and consumption systems [2]. Achieving this objective requires a fundamental transformation in business models, moving away from conventional production and consumption systems to more sustainable alternatives. However, the existing literature lacks clarity and consensus on TBMs and their role in driving sustainability transitions [4,5]. This background sets the stage for the research by reviewing the relevant literature on STT [6], SNM [7], TBMs approach Kavadias and Loch [8], the role of ATs, and the sustainability elements in driving TSBM [9,10].

### 2.1. Sustainability Transitions Theory (STT) in the Start-Ups Business Context

The STT offers valuable support for transformative start-up BMs towards decarbonization. STT is a conceptual framework that aims to understand and ease the shift from unsustainable socio-technical systems to more sustainable alternatives [6]. By applying the STT framework to the context of BMs, researchers can analyze the complex interplay between factors and actors that drive changes in organizational strategies and practices regarding decarbonization.

Start-up BMs, as vehicles for innovation and experimentation, are well-positioned to embrace sustainability principles and contribute to decarbonization efforts. By incorporating the principles of STT, these BMs can be designed and executed with a focus on sustainable practices from the outset [11,12]. This allows them to align their operations and strategies with environmental goals without being bound by legacy systems.

Through the lens of STT, the TBM becomes essential in the perspective of sustainable transition. Start-ups can adopt different types of BMs, such as business-to-business (B2B), business-to-consumer (B2C), and direct-to-consumer (D2C), among others [13], that inte-

grate sustainability into their core strategies. By incorporating technological improvements, start-ups can enhance their organizational development and effectiveness, facilitating the transformation towards more sustainable practices [2,14].

The progress of TBM, which is understood as having the potential to bring about substantial shifts in value creation, alter market dynamics, and revolutionize the process of value capture, holds significance in the realm of sustainability. This progress contributes to addressing environmental and social challenges by means such as carbon emission reduction, the promotion of energy efficiency, encouraging sustainable consumer behavior, and driving innovation [14,15]. By embracing sustainable practices and collaborating with several sectors, organizations such as start-ups can effectively contribute to decarbonization efforts and support the goals of sustainable development [1].

In this context, collaboration for sustainable practices enables the development of TBM to drive sustainable solutions and mitigate environmental and social problems [2]. By anchoring themselves in TSBM, organizations, including start-ups, can reduce their environmental footprint while enhancing their competitiveness through environmentally friendly practices [16]. Collaborations between sectors, enabled by advanced technology sources, can lead to emissions reduction, resource conservation, and the adoption of product-as-a-service and shared-use models [17,18].

Elements such as digital transformation (DT), sharing economy (SE), circular economy (CE) and Sustainable Development Goals (SDGs) are also essential in sustainable transitions and business transformation [19,20]. These concepts offer innovative and efficient solutions to address economic, social, and environmental challenges [2]. By integrating these elements into their models, start-ups can contribute to a more sustainable and resilient economic development that tackles climate change, resource depletion, and social inequality.

Thus, by embracing the principles of the STT, start-up BMs can optimize organizational processes, improve resource efficiency, and accelerate the adoption of sustainable solutions. This transformation can reduce waste, integrate sustainability into core business operations, mitigate economic, social, and environmental impacts, and foster fair and sustainable economic development for all [3,14,21].

*2.2. Strategic Niche Management (SNM) and the Role of Start-Up Business Models in Sustainable Transitions*

SNM theory deals with the emergence and diffusion of innovations within niches, which are protected spaces for experimentation and learning. Within the realm of BMs provides insights into how start-ups and innovative organizations can explore and develop sustainable BMs within their niches, paving the way for broader adoption and diffusion of these transformative models in the market to create protected spaces for experimentation and learning with ATs and practices that can potentially disrupt the dominant regime [7].

In the context of sustainable transitions, start-up BMs can play a crucial role in driving change. These models often focus on addressing environmental challenges and creating sustainable solutions. They have the advantage of being able to align their entire organizational structure and operations with sustainable principles from the outset without being bound by existing infrastructure or legacy systems. Start-ups can embrace sustainable elements such as CE principles, resource efficiency, renewable energy, and other sustainability-oriented practices in their core business strategies [22,23].

Moreover, start-ups can act as agents of change by creating innovative products, services, or technologies that have the potential to disrupt traditional industries and transform them towards sustainability. They can introduce new ways of production, distribution, and consumption that are more environmentally friendly and socially responsible, e.g., Tesla, Envision Solar, Lime Micro-Mobility, etc. Through their entrepreneurial activities, start-ups can demonstrate the viability and scalability of sustainable business models (SBMs), attracting attention and inspiring larger organizations to adopt similar practices [24].

Collaboration with stakeholders like research institutions, investors, and government agencies is common for start-ups, aiding their growth and impact. These collaborations

facilitate knowledge exchange, resource access, and policy support, nurturing environments conducive to sustainable transitions [25].

Start-ups possess the agility, innovation potential, and disruption capability to challenge norms and steer change toward sustainability. Their commitment to sustainable principles, coupled with technological and sustainable elements and novel institutional arrangements, drives the diffusion of SBMs across all sectors [22,24].

### 2.3. Transformative Business Model (TBM) for Decarbonization

The field of TBM is still in its early stages of development. Research conducted on this topic indicates that as of 30 January 2023, a limited number of English articles have been published, with 54 articles identified from Web of Science (WoS) and 85 articles from Scopus.

In the existing body of literature, there remains a lack of clarity regarding the transformative nature of several BMs. The terms "Transformational Business Model" (TBM) and "Business Model Transformation" (BMT) are often used interchangeably. Both concepts have emerged from the original notion of a BM, which posits that a BM should enable organizations to create, deliver, and capture new value [23,26]. This leads to significant changes in market dynamics and benefits both the organization and the consumer [27]. The BMT literature primarily focuses on how organizations identify and exploit opportunities for transformative change while managing the associated risks and challenges of implementing changes in their management systems [27,28]. Therefore, BMT specifically pertains to organizations that already possess BMs and seek to explore innovation changes in their BMs.

For example, Geissdoerfer et al. [29] define BMT as "a combination of its value proposition, value creation and delivery, and value capture elements, as well as the interrelations between the elements and the value network". According to their perspective, BMT refers to an innovation capable of transforming traditional business practices, leading to substantial changes in the way value is created, delivered, and captured in the market. Likewise, the BMT specifically refers to organizations with BMs and that seek to explore innovations that lead to transformative processes.

Consequently, it is imperative to delve into the existing literature concerning transformation approaches within the domain of corporate management. This exploration should encompass contemporary technologies, including AI-ML, digitization, and other relevant aspects (Table 1). Such an investigation aids in comprehending the dynamic nature of corporate management, uncovering insights into its most effective practices.

Additionally, it fosters comprehension of both challenges and opportunities arising in response to technological shifts within the business landscape. Consequently, this understanding becomes a critical asset for organizations aiming not only to thrive in a fiercely competitive environment but also to navigate the swiftly evolving business terrain successfully [30]. On the other hand, BMT denotes the substantial alteration of an existing business model. This process involves modifying various components of the model, including value proposition, mechanisms for value creation and delivery, as well as strategies for capturing value. BMT revolves around reshaping and aligning an organization's current business model to meet emerging market demands, technological advancements, or environmental and social considerations.

A TBM typically incorporates three or more of the subsequent features: (1) Personalization: Offering a more customized product or service that caters to individual and immediate customer requirements by employing technology at competitive prices. (2) Closed-loop process: Minimizing waste, energy consumption, and other costs by reutilizing resources or converting waste into valuable assets. (3) Asset sharing: Facilitating the sharing or rental of idle or underutilized assets, either directly or through a platform. (4) Usage-based pricing: Charging customers based on their usage quantity or nature rather than utilizing a fixed fee structure. (5) Collaborative ecosystem: Creating value by engaging in collaborations with other companies, customers, or stakeholders, often through the utilization of a platform.

(6) Agile and adaptive organization: Demonstrating the capability to swiftly respond to evolving customer needs, market conditions, or technological advancements through a process of experimentation, learning, and iteration [8].

**Table 1.** Transformation insight by different segments.

| Definition | Cor | Tec | Dig |
|---|---|---|---|
| "Digital transformation has been defined as the use of new digital technologies, such as mobile, artificial intelligence, cloud, blockchain, and the Internet of things (IoT) technologies, to enable major business improvements to augment customer experience, streamline operations, or create new business models". [30] | | | √ |
| "The technological revolution that includes digitization, artificial intelligence, big data, and Internet of Things transforms and changes the character of products, processes, markets and competition across the economy". [14,31] | | √ | √ |
| "...Digital transformation signifies the transformational or disruptive implications of digital technologies for businesses and society. Digital transformation is a concept not limited to particularly innovative businesses, digital start-ups, or high-tech giants. It is a process that embraces companies of all sizes, operating in the most diverse industries, as well as their stakeholders". [32,33] | | √ | √ |
| "Artificial intelligence (AI) can replace humans and explain what is important to consider in making the transformation to the digital organization... the increased implementation of electronic services and automation coupled with the general transformation". [33] | | | √ |
| "Digital transformation is a central driver towards innovation and business renewal, especially for established small and medium-sized- enterprises (SMEs) in the contemporary economy". [34] | | | √ |
| "...a new technology or business model that disrupts existing markets and creates new ones". Technology is enabling new business models in the education industry, and how these models are creating new opportunities for learning and growth". [35] | | √ | √ |
| "...a change in how a firm employs digital technologies, to develop a new digital business model that helps to create and appropriate more value for the firm". [36] | | | √ |
| "...business model transformation is straightforward: it is a change in the perceived logic of how value is created by the corporation, when it comes to the value-creating links among the corporation's portfolio of businesses, from one point of time to another". [37] | √ | | |
| "A business model transformation is triggered by market process signals cognized as radical by both unit and corporate level managers". [38] | √ | | |
| "BM transformation includes a complete rethinking of organization's BM elements, resulting into a radically new value proposition". [39] | √ | | |
| "...generating and transforming business models to pursue opportunities—that we have regarded as temporary, unique and driving beneficial change—through timely and uncertain business model transformations, where the business model is described as a combination of Priorities and Capabilities". [40] | √ | | |

Cor—Corporation/Management, Tec—Technology, Dig—Digitalization AI BIG data, IoT.

TBM refers to innovative initiatives that transform existing BMs in the market into disruptive new models. Companies like Airbnb, Uber, Netflix, Spotify, and Zipcar—all considered start-ups at some point in their history—serve as examples of TBM, as they have leveraged ATs to introduce novel value propositions and revenue streams that challenge the traditional economic models within their respective industries.

Briefly, TBM refers to radical changes in the way a business or organization works. It considers people, processes, and technology to make organizations more competitive. So, TBMs are bold, strategic, and seismic changes that incipient organizations such as start-ups make to accelerate change and growth beyond the incremental advances typical of innovations.

### 2.4. Role of Advanced Technologies (ATs) for Transformative Start-Ups Business Models

The incorporation of ATs, such as AI-ML, plays a crucial role in smoothing the start-up' TBMs towards decarbonization. These technologies offer several opportunities to optimize resource utilization, improve energy efficiency, and enable data-driven decision-making for sustainable practices. Understanding the impact of ATs on driving TBMs can provide helpful insights for achieving decarbonization objectives [34].

AI, which refers to the use of algorithms and ML to automate and enhance decision-making processes, possesses significant potential to support organizations in their transition toward sustainability [2,41,42]. The integration of AI-ML, along with other elements, into the production process is fast-evolving. Recent data indicates that AI adoption is more common among innovative organizations aiming to create new products and services, reduce costs, and develop new business practices [2].

Several production sectors leverage AI for purposes such as quality control (identifying defects or deviations in production), predictive maintenance (proactively scheduling maintenance to reduce costs), automation (minimizing bottlenecks waste and improving efficiency), and manufacturing (enhancing the speed and effectiveness of the production system) [2]. Studies have confirmed the advantages of AI adoption, with research revealing that AI represented over 5% of earnings for companies that implemented it in 2019 [2]. Furthermore, companies utilizing AI in their BMs experienced increasing revenue year after year. In 2022, 94% of business leaders regarded AI as critical to enterprise success, with 79% of respondents reporting the deployment of three or more AI types within their organizations in 2021, compared to 62% the previous year [43].

ATs, combined with sustainable elements, offer several benefits for the creation of SBMs. These benefits include cost reduction, production process optimization, increased productivity, enhanced market responsiveness, improved resource efficiency, accelerated management processes, and increased profitability [2]. For instance, the implementation of smart grid management in the energy industry enables the integration of renewable energy sources (RES), replacing fossil fuel consumption and enhancing energy efficiency [44].

AI-ML can empower start-ups by enabling dynamic processing and analysis of vast databases, utilizing ML techniques to extract valuable insights from extensive sets of information, and supporting strategic decision-making. Start-ups can also leverage AI-ML to streamline internal processes, personalize customer communication and experiences, and enhance operational efficiency. The utilization of AI-ML by start-ups can further assist by attracting new investments and differentiating themselves in the market with transformative solutions.

### 2.5. The Elements of Sustainability That Foster the Development of Sustainable Businesses for Start-Ups

Sustainable elements, such as the sharing economy (SE), the circular economy (CE), and the Sustainable Development Goals (SDGs), play a crucial role in supporting start-ups to develop businesses aligned with sustainable transitions [19,20,22,24]. These elements offer innovative and effective approaches to address the economic, social, and environmental challenges of our time. By embracing these principles, start-ups can contribute to the achievement of sustainable development while creating, capturing, and delivering value for their organizations and society.

The SE is a concept that promotes the efficient use and sharing of resources, enabling individuals and businesses to access and utilize assets and services without the need for ownership. Start-ups can leverage the SE model to maximize resource utilization, reduce

waste, and foster collaboration [24]. For example, platforms like Airbnb and Uber have revolutionized the accommodation and transportation sectors by facilitating the sharing of unused spaces and vehicles. By applying the principles of the SE, start-ups can reduce their environmental footprint and promote sustainable production and consumption patterns.

The CE focuses on designing out waste and pollution, keeping products and materials in use for as long as possible, and regenerating natural systems. Start-ups can adopt CE principles, such as reuse, recovery, recycling, and reduction, to transform their BMs and operations [45]. By embracing product-as-a-service models, start-ups can shift from selling products to providing access to product-as-a-service, promoting the reuse and longevity of products. Additionally, incorporating sustainable design and material choices can enable the recovery and recycling of materials at the end of a product's life cycle. By embracing the CE, start-ups can reduce waste, conserve resources, and create innovative solutions that align with sustainability goals.

The SDGs, established by the United Nations, provide a comprehensive framework for addressing global challenges and promoting sustainable development. Start-ups can align their business strategies with the SDGs to contribute to the achievement of specific targets and indicators. For example, a start-up focused on RES can contribute to SDG 7 (Affordable and Clean Energy) by providing clean energy solutions. By integrating SDGs into their BMs, start-ups can demonstrate their commitment to social and environmental responsibility, attract ethical consumers, and access new markets [22].

By incorporating elements of the SE, the CE, and the SDGs into their operations, start-ups can develop businesses that are not only financially viable but also environmentally and socially responsible. These sustainable elements enable start-ups to optimize resource utilization, minimize waste, foster collaboration, and contribute to global sustainability efforts. Furthermore, by embracing these principles, start-ups can transform themselves in the market, attract investment, and build resilient and future-proof businesses. As sustainable development becomes increasingly important, start-ups that prioritize these elements will have a competitive advantage and play a vital role in driving sustainable transitions towards a more sustainable and inclusive future.

## 3. Methodology

To discern the factors that contribute to the TBMs towards sustainability, this study employs a research method that integrates the collection and analysis of data with interviews conducted with knowledgeable professionals hailing from start-up ventures. By combining these approaches, a comprehensive understanding of the driving elements behind the sustainable transformation of BMs can be obtained.

### 3.1. Overview

To assess the essential elements within the BMs of start-ups that instigate changes in production and service delivery methods, thereby facilitating the transition towards sustainability, the methodological approach of this study was divided into three distinct parts (Figure 1).

Initially, we conducted a literature review on ATs, sustainable elements, and TBM in the context of start-ups using the Web of Science (WoS) and Scopus platforms. This review provided the grounds for our study. Subsequently, we employed the Methodological Analysis of Secondary Data (SDMA), drawing upon the elements identified in the literature review. SDMA is a crucial component for the development of the study [46].

Finally, we conducted a series of interviews with start-ups and stakeholders to evaluate the key factors identified during the data collection and analysis phase and get new insights. A qualitative research methodology was employed, utilizing semi-structured techniques in conjunction with the grounded theory analysis. To ensure the effectiveness of the interview process, a pilot study was conducted prior to the real interviews. Stakeholders and a pool of forty-four sustainable start-ups in Portugal, representing 100% of the selected sample of start-ups, comprising approximately 26%, were successfully interviewed. These

approaches allowed a comprehensive examination of the factors influencing the transition of production and service delivery methods in start-ups towards sustainability.

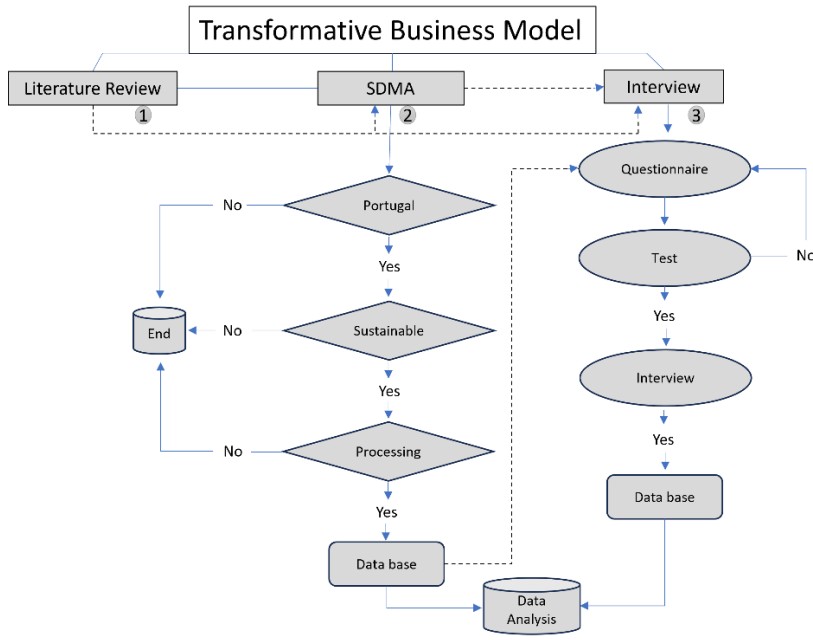

**Figure 1.** Methodological scheme.

### 3.2. Methodology of Empirical Analysis

To identify the essential components of start-up BMs, we conducted an analysis on a dataset consisting of 225 start-ups. We carefully applied exclusion criteria to remove any start-ups that were selected multiple times, resulting in a final selection of 197 start-ups. These start-ups were recognized either through data from the Web of Summit or through awards in Portugal and internationally through internet research covering the years 2014 to 2020.

The selection of start-ups, rather than companies in general, was deliberate due to their characteristic innovative proposals and frequent adoption of AT resources, incorporating new value propositions and transformative BMs. Blank and Dorf [47] emphasize the stark differences between companies and start-ups, stating that "a start-up is a temporary organization in its initial operational stages, founded by one or more entrepreneurs to develop a new product or service while seeking a scalable business model" while a company is "a permanent organization designed to execute a repeatable and scalable BM". Therefore, once a start-up successfully identifies the BM that defines it, it ceases to be a start-up [48]. While start-ups strive for transformative and appealing BMs, companies possess well-defined BMs, with the challenge being their execution and improvement to achieve targeted success.

The Web Summit, considered one of the largest technology conferences in Europe, has received support from the Portuguese government for annual events up to 2028. Portugal was chosen as the focal point of this study due to its recognition of start-ups as potential catalysts for accelerating innovation, driving economic development, and transitioning to a low-carbon society. As of 2022, Portugal boasted 165 incubators and accelerators, 2236 operational start-ups, and 59 active investment programs [49]. The chosen analysis period aligns with the Portuguese government's policy of fostering start-ups, culminating in the decision to host the Web Summit in Lisbon starting in 2014.

Within this study, we explore a TBM, as conceptualized by Kavadias and Loch [8], which possesses the ability to fundamentally reshape operational elements within businesses. This TBM triggers significant shifts in value creation and delivery, inducing substantial transformations in market dynamics and the process of value capture. Therefore, the

TSBM must have the capacity to fundamentally alter existing BMs [9] toward decarbonization methods [50], foster sustainable consumption habits [51], mitigate anthropogenic emissions [52], and enable radical transformations in the production and consumption systems, e.g., Amazon, Google Ads, and Tesla Motors.

### 3.3. Characterization of Empirical Analysis

The characterization of the research aimed to understand the multiple dimensions of start-up TBMs. For this purpose, sample characterization elements were carefully selected which were stored in a database (Table 2) to allow the data to be analyzed from different perspectives.

**Table 2.** Attributes that characterize the database approach.

| Sustainable Elements | Business | Sector | Profile | Root | Evolution | Sustainable Transition | Stage |
|---|---|---|---|---|---|---|---|
| SE<br>CE<br>SDGs | B2C<br>B2B<br>B2G<br>B2B2C<br>D2C<br>B2R | Transport (TP)<br>Building (BL)<br>Consumer goods (CG)<br>Industries (ID)<br>Services (SV)<br>Other<br>(OT) | Limited (LT)<br>Corporation<br>(A.S)<br>Sole<br>proprietorship<br>(SP) | Laboratories (LB)<br>Company (OG)<br>Self-employed<br>(SE) | Active (AC)<br>Sold (SD)<br>Closed (CL) | Improving<br>Shifting<br>Avoiding | Beginner<br>(BG)<br>Senior (SR) |
| Techno elements | | | | | | | |
| AI<br>ML<br>IOT | | | | | | | |

The different types of organization classification adopted by this study are:

(a) Start-up (ST) is considered to be an organization or project undertaken to seek, develop and validate a scalable BM and an organization or project up to 3 years old and has less than 10 employees [53].

(b) National company (NC) is considered to be a start-up with more than 3 years of life that has not advanced into a unicorn but continues to operate in the national market (NC).

(c) Multinational company (MC) refers to a start-up with more than 3 years of existence that has not evolved into a unicorn but which continues to operate in the market with a scope beyond the limits of Portugal (MC).

(d) Unicorn business (UN) is considered to be a privately held start-up company valued at over $1 billion [54].

(e) Spin-off (SO) is considered a new organization that is formed by another organization [55].

The characterization of BMs [13] formed the backbone of the database, enabling a deeper understanding of the dynamics behind start-up BMs. This characterization assists in identifying trends and challenges specific to start-ups, guiding business decisions and adaptations to meet market demands.

The BMs were categorized into (Table 2): (a) B2C, business-to-consumer (transaction between a company and the final consumer), (b) B2B, business-to-business (business transaction between organizations), (c) B2G, business-to-government or NGO (negotiations between companies, associations and governments), (d) B2B2C, business-to-business-to-consumer (transactions between companies aimed at selling to the final customer), (e) D2C, direct-to-consumer (marketing from the manufacturer to the final consumer), (f) B2R, business-to-research or R2B, research-to-business (research to generate business such as financed projects to find sustainable solutions) [13].

The choice of transport, building, and industry sectors, among others (Table 2), was important to identify AT, environmental influences, and business strategies. Together, the

sectors selected to create the start-up database represent around 75% of greenhouse gas (GHG) emissions [56].

Elements to indicate transitions to sustainability were classified into SE, CE, and SDGs. The choice of these three attributes was due to the importance they have for the transition to sustainability. They provide innovative and effective solutions to reduce waste, increase resource efficiency and deliver a more sustainable future [22,24,45].

ATs or techno elements such as AI-ML and its algorithms and IoT can help reduce resource consumption and waste in both production and consumption. AT can help increase the efficiency of resource use by enabling real-time collection, analysis, and decision-making, e.g., smart transportation systems can reduce traffic congestion, save fuel, and decrease air pollution [43].

SE promotes the sharing of resources, assets, and services, which can help reduce waste and overconsumption; e.g., vehicle sharing has the potential to reduce the number of vehicles on the road, increase asset utilization and reduce environmental impact [24]. CE is about minimizing waste and regenerating natural systems by keeping materials and products in use for as long as possible. By embracing CE principles, start-ups can reduce their environmental impact, improve resource efficiency, and create new business opportunities [45]. SDGs are vital for the sustainability of start-ups. They provide a framework for addressing global challenges, offer opportunities for innovation and market growth, attract investment and partnerships, and enhance resilience in the face of uncertainty [22].

The categorization of sustainable transformation elements was based on the mechanisms of improving and shifting consumption and production and avoiding emission [1]. The categorization by this criterion allowed us to understand the type of mitigation proposed by the start-up.

### 3.4. Recruitment of Interviews

The identification of start-ups and relevant stakeholders was carried out based on data collected during the empirical analysis phase, supplemented by extensive internet research. The participants for this study were selected using a purposive sampling method, wherein specific individuals were chosen based on their expertise and relevance to the research objectives. This approach ensured that the participants, including start-ups and stakeholders, possessed valuable insights into sustainable transformations in business models, aligning with the study's focused investigation. The interviews were conducted between the months of April and June 2023, with an average duration of approximately 50 min. The interviews were conducted virtually, covering the entire Portuguese territory.

Respondents were recruited through email and telephone. Each invited individual received a minimum of three communication attempts, including one to forward the invitation letter and at least two additional follow-up contacts. Participants who accepted the interview invitation received a minimum of three additional communications, comprising an email expressing appreciation for accepting the invitation and requesting information regarding interview preferences, another to confirm the interview appointment and provide the interview connection link, and finally, a reminder on the eve of the interview.

During the preparatory phase of the study, it was anticipated that the number of interviews would fall within the range of 15% to 35% of the total number of invitees. Consequently, the participation rate of 26% within the sample, given the busy nature of the professionals involved, proves to be highly representative and consistent with other qualitative studies [11,23,57–59].

### 3.5. Sample Characterization of Interviews

Around 26% of the attendees, predominantly start-ups (94% of the sample), were granted an interview. The majority of the interviewees (65%) were in the northern region of the country, while the central region accounted for the second-highest representation (30%). Roughly 60% of the start-ups were classified as small companies, employing up to nine individuals (Appendix B).

Medium-sized companies, with workforces ranging from 10 to 49 employees, constituted approximately 30% of the sample, while large companies with 100 or more employees represented around 10%. The cities of Porto and Lisbon emerged as the favored locations for these start-ups. About 75% of the respondents were male, and more than 90% held the position of Chief Executive Officer (CEO). Among the CEOs, around 70% fell into the middle-aged category (31 to 50 years old), while approximately 15% were juniors (up to 30 years old), and another 15% were seniors (over 50 years old). In summary, the respondents can be characterized as middle-aged male CEOs of small start-ups located in the northern region of Portugal (Table 3).

**Table 3.** Profile of respondents.

| Type of Organization | Region | Org. Size | Title | Gender | Age |
| --- | --- | --- | --- | --- | --- |
| Start-up | Center | Small | CEO | Male | Senior |
| Start-up | Center | Large | CEO | Male | Middle-aged |
| Start-up | North | Small | CEO | Female | Middle-aged |
| Start-up | North | Medium | CEO | Male | Middle-aged |
| Start-up | North | Small | CEO | Female | Middle-aged |
| Start-up | North | Small | CEO | Male | Junior |
| Start-up | North | Small | CEO | Male | Middle-aged |
| Start-up | North | Small | CEO | Male | Middle-aged |
| Start-up | Center | Small | CEO | Male | Junior |
| Start-up | North | Medium | CEO | Male | Middle-aged |
| Incubators/Accelerators | North | Medium | CEO | Female | Middle-aged |
| Other supporting entities | Center | Medium | Other | Female | Middle-aged |

### 3.6. Questionnaire of Interviews

The themes and structure of the questionnaire for the interviews were derived from the key elements identified in the analysis of the start-up database collected during the empirical analysis phase. The questionnaire encompassed a combination of scaled and open-ended questions, and it was divided into three sections: (1) Sociodemographic Characteristics, which aimed to gather relevant data; (2) Examination of experience in digital technology transformation and BMs; and (3) Assessment of sustainable and unsustainable elements in the context of transformational production and consumption.

All sections were designed to corroborate the identified transformative elements of BMs from the empirical analysis phase while also exploring novel insights regarding the potential of BMs to foster sustainability. To quantitatively assess each aspect, a Likert-scale point system [60] was employed. Participants were instructed to rate the items on a scale of one to five, with one indicating a low level of challenge or importance and five indicating a high level of challenge or importance, as perceived by the respondent.

### 3.7. Interview Procedure

The interviews were conducted by a single interviewer to minimize the usual variability associated with multiple interviewers. Following an initial introduction to confirm the purpose of the interview and provide clarifications regarding its procedures and components, the researcher proceeded to gather personal information from the interviewees. This included their name, position, and age, as well as details pertaining to their organization, such as type, size, location, and sector of activity. These data were crucial for comprehending the sociodemographic characteristics of the interviewees.

Subsequently, the interviewees were presented with specific questions aimed at evaluating the elements that have the potential to bring about transformative changes in the

production and consumption of goods and services, with a focus on promoting sustainability. Open-ended questions were included at the conclusion of each section to maintain consistency with the respective topic under examination.

### 3.8. Grounded Theory Analysis

Grounded Theory is a systematic and inductive method used in qualitative research to develop hypotheses and theories based on empirical evidence. It allows for a deeper understanding of phenomena by exploring social relationships and group behaviors, leading to new insights and the development of novel elements [61]. The approach offers benefits such as going beyond surface-level observations, allowing the organic emergence of underlying questions, and providing flexibility to refine concepts and theories as new data emerges [62].

However, there are drawbacks to Grounded Theory. It requires a significant time investment for comprehensive data collection from multiple perspectives, which can be overwhelming for novice researchers. The method may generate large amounts of data, posing challenges in data management. Subjectivity and bias in data interpretation are potential limitations, as researchers' preconceived notions may influence findings, requiring awareness and mitigation of biases [63]. Despite these limitations, Grounded Theory is widely used in several scientific fields, including business, due to its rigorous and systematic approach to data analysis, enabling a comprehensive understanding of complex phenomena [64].

The purpose of combining grounded theory analysis with the Likert scale was to provide validation or corroboration for the qualitative findings. During the qualitative analysis, specific themes or categories were identified for further exploration, and the Likert scale was utilized to examine the responses about these areas. By incorporating the Likert scale into grounded theory analysis, additional support was obtained to reinforce the qualitative findings.

## 4. Results

### 4.1. Secondary Data Analysis

The characterization of the empirical analysis reveals that around 15% of start-ups can be classified as sustainable. Among these sustainable start-ups, half (50%) are engaged in activities related to AT and incorporate sustainable elements.

Furthermore, over 60% of start-ups utilizing AI-ML technology follow the B2B2C model. Around 80% of sustainable start-ups adopt either the B2B or B2B2C models. Additionally, more than 80% of all start-ups are associated with the service sector. The majority, around 85%, of start-ups are registered as limited liability companies (LTD), and half of them originate from the business environment. Perhaps most strikingly, every single one of the sustainable start-ups under study has consciously integrated one or more principles from the Sustainable Development Goals (SDGs) into their day-to-day operations. This highlights a remarkable dedication to contributing to the global sustainability agenda, as outlined by the United Nations. By weaving these principles into their business models, these start-ups are tangibly aligning their strategies with broader societal and environmental aspirations.

Start-ups are not only adopting AI-ML to enhance the development of products and services, as discussed in the literature on the benefits of digitalization in production and consumption [2]. According to existing research, AI-ML solutions are primarily utilized by B2B and B2C models due to their cost-effectiveness and fast response time.

However, our analysis demonstrates that AI-MLs are increasingly attracting attention (Figure 2) for transforming start-up BMs. This transformation goes beyond the boundaries of production and commercialization and extends to new arrangements involving controls and stakeholder relationships. Specifically, our findings highlight the growing interest in AI-ML within the B2B2C model, which combines elements from both B2B and B2C models, creating a more complex logic.

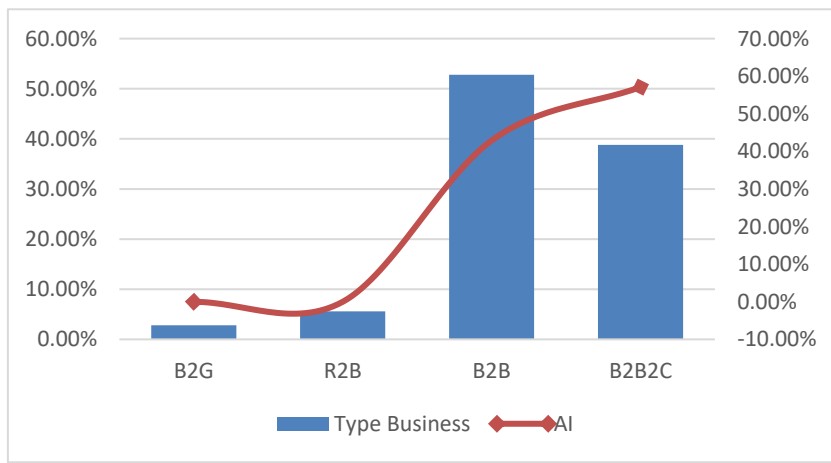

**Figure 2.** Start-up business model characterization.

Hybrid models like B2B2C necessitate increased interaction, controls, quick decision-making, rapid responses, and cost reduction to minimize the lead time from production to consumption and ensure viability. Thus, AI-ML has become an indispensable tool for start-ups, particularly those adopting hybrid models. It empowers them to differentiate themselves and establish new arrangements, providing a significant long-term competitive advantage.

This trend can be attributed to the fact that over 50% of sustainable start-ups are associated with sectors such as AT, SE, and CE. Ultimately, by reengineering traditional BMs, start-ups can devise new, innovative, and sustainable approaches to meet customer needs, create value, and capture that value through revenue. This can pave the way for sustainable growth and success (more information on data analysis in Appendices A and B).

Prioritizing Production and Consumption in Sustainable Start-Ups: Trends and Challenges

Approximately 60% of sustainable start-ups prioritize improving production and consumption, while around 30% focus on transforming their production and consumption methods. A smaller portion, about 10%, specifically concentrates on avoiding emissions (Figure 3a).

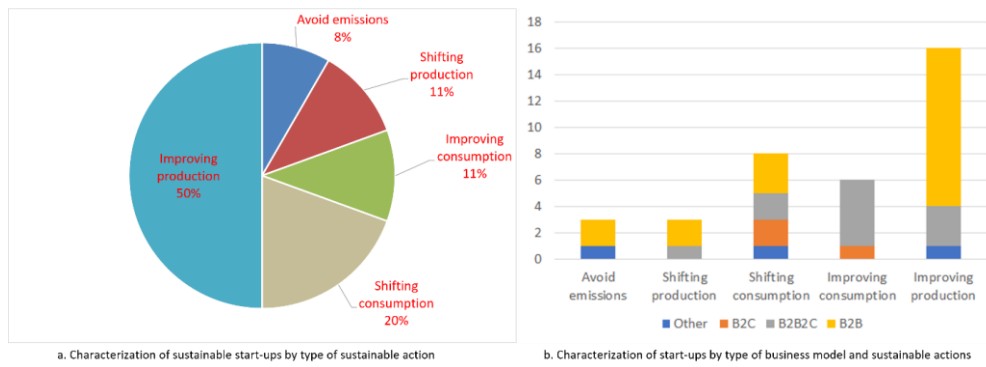

a. Characterization of sustainable start-ups by type of sustainable action

b. Characterization of start-ups by type of business model and sustainable actions

**Figure 3.** Mechanisms for enhancing decarbonization and sustainability.

The higher inclination towards enhancing production and consumption can be attributed to its practicality and profitability. By improving efficiency, reducing waste, ensuring product quality, and meeting consumer demands, start-ups can achieve tangible benefits. This approach is often more accessible and yields positive results in terms of resource optimization and customer satisfaction.

Conversely, the lower preference for shifting the production and consumption system stems from the significant costs involved in such transformations. Overhauling assembly

lines or changing production sites necessitates substantial investments in machinery and equipment, the development of new processes, and personnel training. These financial implications act as a deterrent for start-ups, leading to a relatively lower adoption rate of this approach.

Furthermore, the limited interest, accounting for approximately 10%, in emissions avoidance can be attributed to the high costs and limited development of equipment capable of effectively mitigating carbon emissions. Technologies such as CCUS and CCS, which aim to prevent carbon emissions, are still in the incipient stages of development and often come with considerable financial burdens.

The analysis of start-ups based on their BM type (Figure 3b) reveals that the B2B model primarily focuses on enhancing production. The underlying notion is that if an industry offers a high-quality and cost-effective product, it will have a higher chance of success. Conversely, the B2B2C model is more inclined towards improving consumption, likely due to the prevalence of start-ups in the services sector. From an emission perspective, the B2B and B2G models are predominantly observed. Currently, AT, such as carbon capture and use or storage (CCUS) and carbon capture and storage (CCS), is not accessible for the B2C or B2C2B models.

The cross-data analysis to determine the type of BM by sector (Figure 3b) and maturity level of projects by the start-up phase revealed a greater share of newly created projects covering the services and industrial sectors. The B2B model, which constitutes approximately 50% of sustainable start-ups, was found to be more prevalent in the sustainable industry and service sectors. Conversely, the B2B2C model, which constitutes approximately 40%, was found to be more widespread among the other sectors. This may indicate a growing interest among start-ups in a TBM that enables sales to both companies and consumers.

Ultimately, the sector that a start-up focuses on will depend on several factors, including market opportunity, the founder's expertise and experience, and the available resources and support. As widely known, the service sector is important for Portugal, so it seems natural for Portuguese start-ups to opt for the provision of services. However, the start-ups moving towards a hybrid model may be linked to the success of several companies, including prominent chip manufacturers such as Intel, Snapdragon, and Qualcomm, that adopted the B2B2C model, as well as other segments that have been showing success with the B2B2C model, such as Amazon, Alibaba, Shopify, App Store and Google Play. By reinventing traditional BMs, start-ups can find new and transformative ways to meet customer needs, generate value and capture revenue, leading to greater success and growth.

### 4.2. Interviews Results

The results of the interviews highlight a clear connection between BMs and ATs when striving for sustainable value creation. This supports the conclusions drawn from the empirical analysis. Moreover, they highlight that the integration of technological resources is primarily driven by the desire to achieve environmental objectives, as these resources play a pivotal role in facilitating the transformation of BMs towards sustainability.

### 4.2.1. Advanced Technological Resources, Transformation of Business Model and Sustainability Transitions

Figure 4a presents the results of the interviews. More than 60% of respondents reported utilizing AI, while approximately 20% stated that they are currently in the process of implementing AI-ML. Half of the participants acknowledged using AI-ML specifically for production purposes, while the other half employed it more broadly for tasks such as controls, customer interactions, supplier management, and more.

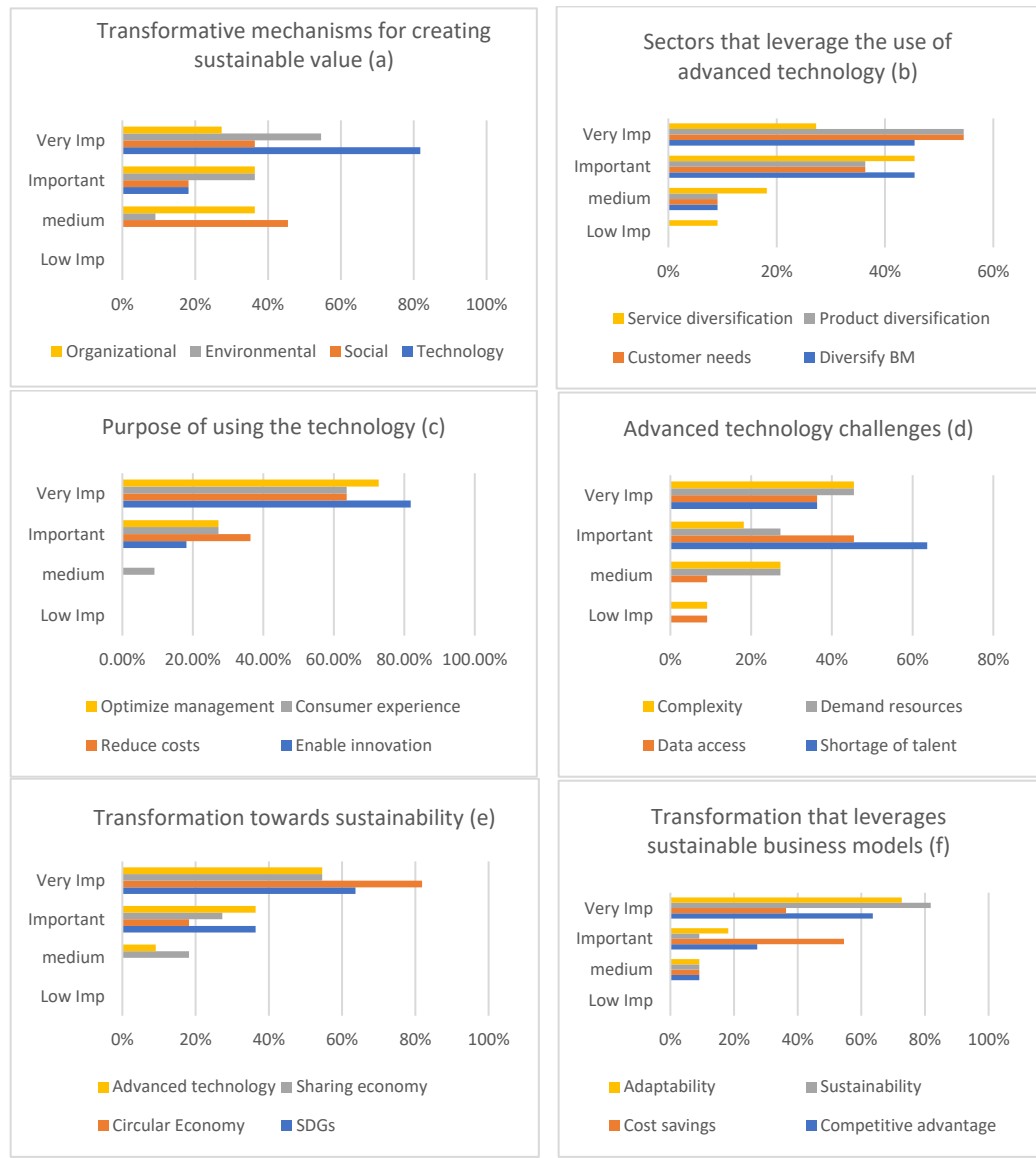

**Figure 4.** Analysis of technological and sustainable elements in business models.

Less than 20% of start-ups indicated a lack of adoption of this technology. However, they also emphasize that technology alone is insufficient to drive TBM toward sustainability. Instead, a comprehensive approach is required, wherein ATs are coupled with a range of sustainable components managed through novel institutional arrangements (Figure 4a).

Besides that, the data presented in Figure 4a provide evidence that start-ups are directing their attention both to socio-environmental aspects and to elements of advanced technology to drive substantial transformations in the creation and achievement of sustainable value. In doing so, they are navigating a delicate balance that allows for the transformation of their business models towards low-carbon socio-economic development (Figure 4a).

Most respondents acknowledge that the implementation of AI-ML solutions necessitates specialized technical expertise. Consequently, there is a need to invest in the cultivation of relevant skills and talents to ensure the successful deployment of ATs. The management of technologies and other organizational resources, in conjunction with social concerns, significantly influences the outcomes of sustainability transitions.

The adoption of AT to diversify production and consumption and cater to consumer needs (Figure 4b) is widely regarded as a crucial pathway toward achieving more sus-

tainable practices, according to the interviewees. By implementing effective institutional arrangements to manage AT and available resources, opportunities for innovation arise, leading to the introduction of novel product offerings and enabling quicker market entry with customized solutions (Figure 4c). However, transforming BMs towards decarbonization presents significant challenges. Addressing the shortage of skilled professionals proficient in ATs like AI-ML remains crucial despite the advancement of remote work, which expands the recruitment possibilities for such professionals. Additionally, overcoming obstacles related to accessing large volumes of high-quality data (Figure 4d) and developing the capacity to effectively manage such data are vital for substantial progress in the transition towards more sustainable practices.

Amidst these dynamics, start-ups are strategically positioned to explore and pioneer the development of transformative sustainable business models within their respective niches. These niches serve as fertile grounds where innovation is nurtured and sustainability-oriented practices are incubated. For instance, start-ups focusing on AI-ML technology might design business models that integrate renewable energy solutions into their operations, minimizing carbon footprints. Such innovation resonates within their niches, setting an example for others. These niche innovations have the potential to ripple beyond their boundaries, influencing and inspiring larger players across several industries. As more start-ups successfully demonstrate the viability of these sustainability-driven models, they propel the broader market toward embracing more eco-conscious practices. Consequently, a collective movement is triggered, cascading the positive impacts of sustainability-focused niche innovations across diverse sectors, contributing to a more sustainable and resilient global business landscape.

### 4.2.2. Sustainable Elements in the Transformation of Production and Consumption

In terms of transforming production and consumption towards more sustainable modes, respondents highlighted the significance of sustainable elements such as CE and SDGs, which were identified as crucial factors. The SE was identified as the third most important element (Figure 4e). AT was highlighted as interviewees recognized its potential to enhance the efficiency of production processes, leading to reduced energy and raw material consumption. When considering operational factors that optimize production and consumption practices, start-ups emphasized the importance of the pursuit of sustainability as a key driver for transformation. By minimizing waste, conserving resources, and adopting sustainable practices, start-ups can contribute to decarbonization and enhance their corporate reputation. Additionally, adaptability to market conditions and consumer needs emerged as the second most important aspect (Figure 4f). Respondents believed that by being agile and responsive to emerging trends and technologies, start-ups can proactively stay ahead of the curve in a rapidly evolving business landscape.

### 4.2.3. The Influence of Technology on Business Models and Other Insights

The interviews provide further evidence supporting the empirical analysis, confirming that the transformation of BM towards sustainability is driven by key elements such as automation and algorithms. The incorporation of these elements, alongside one or more sustainable components, reveals a growing trend towards hybrid BMs.

The consumer-to-business (C2B) model emerges as particularly promising for decarbonization efforts. Under this approach, consumers are incentivized to return goods instead of discarding them and may even receive compensation. Many businesses with C2B principles are currently being developed. During one interview, a start-up was mentioned that collects electronic equipment and remunerates individuals who return used items. Thanks to AI-ML, this process can be scaled effectively.

There is a shift in the logic of BMs dominated by B2B and B2C, where consumers traditionally sought out suppliers (physically or virtually) for their purchases. With the AT, such as AI-ML, a new logic is emerging where suppliers proactively find consumers. For instance, when a consumer searches for a laptop on a digital device (such as a smartphone,

tablet, or computer), algorithms rapidly locate the consumer and enable suppliers to advertise laptops whenever the consumer is online. These algorithms can also provide guidance on how to dispose of the used equipment, generating income for all parties involved and relieving the consumer of the disposal task. This transformation brings multiple benefits to the economy, society, and the environment.

### 4.2.4. Start-Ups and the New Institutional Arrangements

Start-ups are leveraging advanced technologies and sustainable elements to revolutionize their BMs, aiming to decarbonize products and services and expand into new markets, adapting to societal yearnings for low-carbon development. As a result of these transformations in production and supply, the development of new institutional arrangements becomes essential to integrate transformation into the DNA of the business models.

These institutional arrangements play a vital role in fostering collaboration, influencing regulations, setting industry standards, attracting investment, promoting knowledge sharing, facilitating ecosystem collaborations, and providing a smarter use of available resources. By creating an enabling environment, these new institutional arrangements, acting in concert with TAs and sustainable elements, provide TBM and enable startups to make broader contributions to sustainable practices in the business landscape (Figure 5).

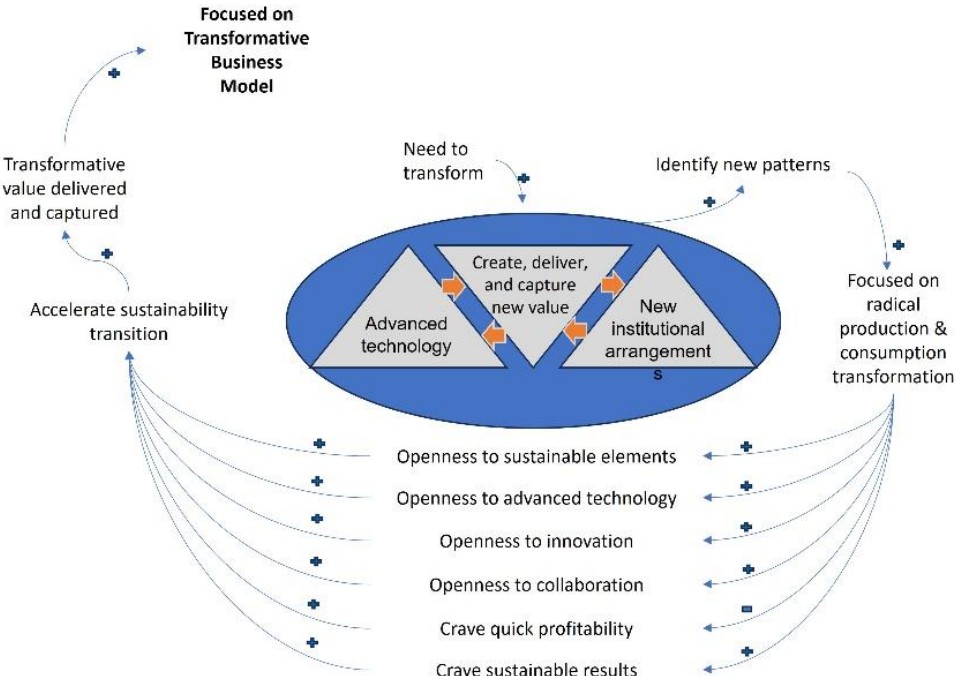

**Figure 5.** The system dynamics behind the transformative business models towards sustainability.

### 5. Conclusions

To identify key components of start-up BMs that enable the shift to more sustainable production and delivery modes and contribute to the rapid decarbonization of several economy sectors, this study analyzed a dataset of start-up companies and conducted stakeholder interviews.

This study's primary achievement was the identification of crucial elements that promote the transition to more sustainable modes. These elements encompass the integration of advanced technological systems with sustainable elements that interact harmoniously within new institutional frameworks, enabling smarter management of available resources to improve practices to decarbonize production and consumption, transform business models, and contribute to sustainability goals. Among these sustainable elements (e.g., reuse, recovery, recycling, and reduction) supplemented by sharing systems elements, SDGs hold significant transformative potential, particularly in the realm of environmental

conservation. The cooperation of these elements with AI-ML algorithms expands the conditions for the transformation of BMs, further amplifying their impact within the context of evolving circularities. Furthermore, the study highlighted the potential of C2B and B2B2C models in supporting sustainable transition pathways.

The results of this study also shed light on several relevant aspects of the transformation of BMs for sustainability in start-ups. The empirical base analysis revealed that sustainable start-ups prioritize the integration of advanced technological systems and sustainable elements in their BMs. Notably, over 60% of start-ups utilizing AT, such as AI-ML, follow the B2B2C model, indicating a shift towards more integrated BMs. Furthermore, more than 80% of all start-ups operate in the service sector. It is worth noting that all start-ups integrate one or more SDG principles into their operations.

Interviews conducted with stakeholders further prove the importance of advanced technological resources in driving the BMs' transformation towards sustainability. The adoption of AI-ML tools was recognized as crucial to increasing the efficiency of production processes and reducing energy and raw material consumption. However, it was highlighted that technology alone is insufficient and must be accompanied by sustainable elements and managed through new institutional arrangements capable of adapting their business strategy to a new socio-economic and eco-environment reality.

The study also stresses the emergence of the C2B model as a promising approach to decarbonization. This model encourages consumers to return merchandise rather than discard it, creating opportunities for companies to adopt circularity practices. In addition, the integration of AI-ML and its algorithms allows suppliers to proactively find consumers and offer personalized product recommendations, as well as the option of proper disposal of used items.

The findings emphasize the evolving landscape of BMs in start-ups, where the integration of AT, sustainable elements, and new institutional arrangements are key drivers of sustainable transitions. By reshaping traditional BMs and adopting transformative approaches, start-ups can contribute to decarbonization by creating, delivering, and capturing sustainable value in their BMs while striking a balance that allows for effective capitalization, all in support of the acceleration of sustainability transition pathways.

This study expands the literature on BMs by providing new information to support organizations such as start-ups in their efforts to decarbonize production and consumption systems. The research is also relevant for policymakers who seek to encourage the decarbonization of the economic sectors by identifying the levers that can be used to drive the transformation of BMs towards sustainability.

### 5.1. Limitations

During this study, some limitations were encountered, which should be considered when interpreting the findings. One of the limitations was the low level of detail available on the BMs of the start-up companies included in our dataset. This limitation affected the depth of our analysis and may have influenced the comprehensiveness of the identified elements driving sustainability transitions in BMs.

### 5.2. Recommendations for Future Studies

To address these limitations and further advance our understanding of the transition to sustainable business models in start-ups, we offer the following recommendations for future research: (A) Broader Geographic and Sectoral Scope: Future studies should consider expanding the geographic and sectoral scope of the analysis. This broader perspective could lead to a more representative sample and a more comprehensive understanding of the factors influencing the transformative BMs across several regions and industries. (B) Utilization of AI-ML Tools: Leveraging advanced tools such as AI-ML for the analysis of a substantial sample of start-ups may provide deeper insights into the impact of technology on transformative BM. AI-ML can help identify patterns, trends, and correlations that may not be immediately apparent through manual analysis. (C) Exploration of Additional

Influencing Elements: Future investigations should delve into identifying and exploring other elements, in conjunction with AI-ML that can guide TBMs toward sustainability. This expanded focus may reveal new drivers and strategies for sustainable transformations within the start-up ecosystem.

By addressing these limitations and pursuing these recommendations, future research endeavors can build upon the foundation laid by this study, contributing to a more comprehensive and nuanced understanding of how start-ups can drive sustainability transitions through their business models. Such insights are crucial for both start-up organizations seeking to decarbonize production and consumption systems and policymakers aiming to facilitate sustainable transformations in the broader economic sectors.

**Author Contributions:** Conceptualization, E.C., M.F. and N.B.; Methodology, E.C.; Formal analysis, E.C.; Investigation, E.C.; Resources, E.C.; Data curation, E.C.; Writing—original draft, E.C.; Writing—review & editing, E.C., M.F. and N.B.; Supervision, M.F. and N.B. All authors have read and agreed to the published version of the manuscript.

**Funding:** This research was funded by Fundação para a Ciência e a Tecnologia (FCT) (project ref. PTDC/GES-AMB/0934/2020).

**Institutional Review Board Statement:** Not applicable.

**Informed Consent Statement:** Not applicable.

**Data Availability Statement:** The study data is available upon request from the authors.

**Acknowledgments:** We are grateful for the support received from the Fundação para a Ciência e a Tecnologia (FCT) (project ref. PTDC/GES-AMB/0934/2020). We also thank our colleagues, especially Ana Patrícia Veloso, António Duarte, Joana Sousa, and Tiago Alves, for their contributions. We reserve a heartfelt thank you for the consultant and Ron Boschma for the constructive criticisms shared throughout this endeavor.

**Conflicts of Interest:** The authors declare no conflict of interest.

## Appendix A

**Table A1.** Interview characterizations.

| Organization | Type Organization | | | Organization Size | | | Location | | | | |
|---|---|---|---|---|---|---|---|---|---|---|---|
| | Start-ups | Incub/Acc | Other | Up to 9 | 10 to 49 | 50 to 99 | North | South | Center | Açores | Madeira |
| Start-up #1 | x | | | x | | | | | x | | |
| Start-up #2 | x | | | | | x | | | x | | |
| Start-up #3 | x | | | x | | | x | | | | |
| Start-up #4 | x | | | | x | | x | | | | |
| Start-up #5 | x | | | x | | | x | | | | |
| Start-up #6 | x | | | x | | | x | | | | |
| Start-up #7 | x | | | x | | | x | | | | |
| Start-up #8 | x | | | x | | | x | | | | |
| Start-up #9 | x | | | x | | | | | x | | |
| Start-up #10 | x | | | | x | | x | | | | |
| Incubators/Acc. | | x | | | x | | x | | | | |
| Government body | | | x | | x | | | | x | | |
| | 10 | 1 | 1 | 7 | 4 | 1 | 8 | 0 | 4 | 0 | 0 |
| | 83.3% | 8.3% | 8.3% | 58.3% | 33.3% | 8.33% | 66.7% | 0.00% | 33.3% | 0.00% | 0.00% |

**Table A2.** Interview characterizations.

| Organization | Position | | | | Age | | | | | Sex | |
|---|---|---|---|---|---|---|---|---|---|---|---|
| | CEO | Director | Leader | Other | Up to 20 | 21 to 30 | 31 to 40 | 41 to 50 | Over 50 | Masc. | Fem. |
| Start-up #1 | x | | | | | | | | x | x | |
| Start-up #2 | x | | | | | | x | | | x | |
| Start-up #3 | x | | | | | | | x | | | x |
| Start-up #4 | x | | | | | | x | | | x | |
| Start-up #5 | x | | | | | | x | | | | x |
| Start-up #6 | x | | | | | x | | | | x | |
| Start-up #7 | x | | | | | | x | | | x | |
| Start-up #8 | x | | | | | | x | | | x | |
| Start-up #9 | x | | | | | x | | | | x | |
| Start-up #10 | x | | | | | | x | | | x | |
| Incubators/Acc. | x | | | | | | x | | | | x |
| Government body | | | | x | | | x | | | | x |
| | 11 | 0 | 0 | 1 | 0 | 2 | 8 | 1 | 1 | 8 | 4 |
| | 91% | 0.00% | 0.00% | 8.3% | 0.00% | 16.7% | 66.7% | 8.3% | 8.3% | 66.% | 33.3% |

**Appendix B**

**Table A3.** Characterization of the empirical analysis.

| By Type of Society | Number | % | Sample Size | Number | % |
|---|---|---|---|---|---|
| Limited liability company | 28 | 83.0% | Sample size | 231 | 100.0% |
| Anonymous society | 3 | 8.0% | Sustainable | 34 | 16.0% |
| Sole proprietorship | 2 | 6.0% | Unsustainable | 197 | 84.0% |
| Association | 1 | 3.0% | | | |

**Table A4.** Characterization by type of business model and sectors.

| By Type of Business and Sector | | | | | |
|---|---|---|---|---|---|
| Sector/Type of Business | B2B | B2C | B2B2C | Others | Total |
| Consumer goods (CG) | 0 | 0 | 0 | 0 | 0 |
| Transport (TP) | 1 | 1 | 1 | 0 | 3 |
| Industry (ID) | 4 | 0 | 4 | 0 | 8 |
| Service (SV) | 15 | 2 | 3 | 2 | 22 |
| Other (OT) | 0 | 0 | 1 | 0 | 1 |

**Table A5.** Characterization of the empirical analysis.

| Type of Corporation | Number | % | By Founding Year | Number | % |
|---|---|---|---|---|---|
| Multinational Corporation (MC) | 2 | 5.9% | Y2014 | 5 | 5.5% |
| National Corporation (NC) | 1 | 2.9% | Y2015 | 4 | 8.3% |
| Start-up (ST) | 18 | 52.9% | Y2016 | 2 | 11.2% |
| Scale-up (SU) | 4 | 11.8% | Y2017 | 2 | 13.9% |
| Unicorn (UN) | 1 | 2.9% | Y2018 | 8 | 16.7% |
| Spin-off (SO) | 7 | 20.6% | Y2019 | 5 | 22.2% |
| Other (OT) | 1 | 2.9% | Y2020 | 8 | 22.2% |
| **Working sector** | Number | % | **By conception** | Number | % |
| Transport (TP) | 3 | 8.8% | Scholar | 14 | 42.0% |
| Thermal comfort (TC) | 0 | 0.0% | Executive Business | 3 | 8.0% |
| Consumer goods (CG) | 1 | 2.9% | Entrepreneur/liberal professional/self-employed | 17 | 50.0% |
| Industry (ID) | 8 | 23.5% | **Current stage** | | |
| Service (SV) | 22 | 64.7% | Beginner | 25 | 72.0% |
| Other (OT) | 0 | 0.0% | Experienced | 9 | 28.0% |

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
