# Peer review of "Transformative Business Models for Decarbonization: Insights from Prize-Winning Start-Ups at the Web Summit"

_sustainability, doi:10.3390/su151814007_

Round 1

Reviewer 1 Report

My observations and recommendations.

1. The paper needs to be more concise and straight to the point. Lengthy descriptions can be shortened without losing the essence, making the content more reader-friendly. The  conceptual framework is majority of the paper.

2. Methodological Concerns: There is a level of ambiguity concerning the research methodology:

    Data Collection: The paper mentions interviews were conducted and also claims secondary data was used. However, it's unclear how these interviews were chosen and conducted. What was the sampling method? The sample is quite small and cannot merit valid conclusions

    Strategy: There's a need for clarity in terms of the main data collection strategy. Were the interviews structured, semi-structured, or unstructured? If secondary data was used, from what sources was this data collected, and how was it integrated with primary data? Why both methods applied, is it even neccessary, i dont think the two methods neccessarily fulfill main aim of triangulation.

3. The discussion is not connected to previous research in the field. The authors should link their findings with prior research, thereby placing their work in the broader context of the literature. Also consider adding some research that could enhance background such as Xing, Z., Huang, J., & Wang, J. (2023): "Unleashing the potential: Exploring the nexus between low-carbon digital economy and regional economic-social development in China." And Zhang, Z., Hao, L., Linghu, Y., & Yi, H. (2023): "Research on the energy poverty reduction effects of green finance in the context of economic policy uncertainty."

 "Clustered institutional investors, shared ESG preferences and low-carbon innovation in family firm;, Zhao, S., Zhang, L., An, H., Peng, L., Zhou, H.,... Hu, F. (2023): "Has China's low-carbon strategy pushed forward the digital transformation of manufacturing enterprises? Evidence from the low-carbon city pilot policy."

4. The scientific rigor of the study appears uncertain. The paper combines various literature streams, like Sustainability Transitions Theory (STT), Strategic Niche Management (SNM), and the Business Models approach. However, the integration of these streams should be carried out more systematically. The application of these theoretical frameworks should be more apparent in the analysis and findings.

5.  Are theoretical contributions evident?

6. Most of the tables and figures report on issues that do not require individual visual representation, so the tables should be merged, and figures.

proofreading should be done

Author Response

Dear reviewer.

We sincerely thank you very much for your thoughtful review. Your feedback has been invaluable to us, and we greatly appreciate the insights you've provided. Incorporating your constructive comments, we have revised the article. For your convenience, all revisions have been highlighted in blue. We believe these modifications address the specific areas you highlighted and enhance the overall quality of the manuscript. We are optimistic that these revisions will contribute to the successful publication of our article. Below, you will find your recommendations along with our corresponding responses:

  1. “The paper needs to be more concise and straight to the point. Lengthy descriptions can be shortened without losing the essence, making the content more reader-friendly. The  conceptual framework is majority of the paper”.

Authors (AU): Thank you very much for your comments. We appreciate your feedback. Based on your comments, we have revised some of the text to enhance its clarity and maintain consistency.

  1. “Methodological Concerns: There is a level of ambiguity concerning the research methodology”:

AU: Thank you very much for your comments. We greatly appreciate your thoughtful consideration. We kindly request permission to express a differing viewpoint regarding the arguments presented. The methodology presented offers an elucidation of the research endeavor aimed at comprehending the factors that contribute to sustainable transformations within start-up business models. The methodology encompasses a series of stages such as Research Objectives and Approach, Data Collection, SDMA, Interviews, Sample and Selection, Start-up Characteristics, Business Model Categorization, Sustainable Transformation Elements, Interview Recruitment and Characterization, Questionnaire and Interviews, and Grounded Theory Analysis.

The provided text furnishes an all-encompassing synopsis of the research methodology, meticulously delineating each step undertaken in the investigation of factors influencing sustainable transformations within start-up business models. The incorporation of both qualitative and quantitative methods, as advocated by several authoritative sources, enhances the depth of understanding of the subject matter. Your contributions, along with those of other esteemed authors, substantially enrich the scholarly discourse.

Data Collection: The paper mentions interviews were conducted and also claims secondary data was used. However, A) it's unclear how these interviews were chosen and conducted. B) What was the sampling method? C) The sample is quite small and cannot merit valid conclusions.

PR - A) “it's unclear how these interviews were chosen and conducted”.

AU: Thank you very much for your comments. All aspects of how the interviews were chosen and conducted are highlighted in the methodology sections as follows:

1) Identification of Participants: See sub-section 3.4. of the methodology.

2) Recruitment Process: See Recruitment of interviews sub-section (Section 3.4).

3) Participant Inclusion Criteria: See Recruitment of interviews sub-section (Section 3.4).

4) Interview Timing and Format: See Recruitment of Interviews and Interview Procedure sub-sections (Sections 3.4 and 3.7).

5) Interviewer: See Interview Procedure sub-section (Section 3.7).

6) Interview Structure: See Questionnaire of interviews sub-section (Section 3.6).

7) Data Gathering: See Interview Procedure sub-section (Section 3.7).

8) Questionnaire and Likert Scale: See Questionnaire of Interviews and Grounded theory analysis sub-sections (Sections 3.6 and 3.8).

9) Grounded Theory Analysis: See Grounded theory analysis sub-section (Section 3.8).

10) Validation: See Grounded theory analysis sub-section (Section 3.8).

PR - B) “What was the sampling method”?

AU: Thank you very much for your comments. We sincerely appreciate your valuable feedback. We would like to highlight that the participants for this study were chosen through a purposive sampling approach. Likewise, we have included detailed information about this sampling method in the initial paragraph of sub-topic 3.4 for clarity and transparency.

PR - C) “The sample is quite small and cannot merit valid conclusions”.

AU: Thank you very much for your comments. The sample size of 197 start-ups included in this study, while not large in a quantitative sense, is not inherently indicative of an inability to draw valid and meaningful conclusions. Qualitative research, such as that conducted through interviews and grounded theory analysis, places emphasis on the depth of insights gained rather than statistical significance. Through purposive sampling, specific participants were chosen, ensuring a focused and comprehensive exploration of the factors driving sustainable transformations in business models. The careful selection of diverse start-ups and stakeholders, along with the employment of qualitative methodologies, enhances the richness of the data collected. Moreover, if data saturation was achieved—meaning that further interviews would unlikely yield new insights—the conclusions drawn from this sample can offer valuable and contextually nuanced findings. The qualitative approach employed in this study aligns with the aim of delving into complex issues and understanding the intricate dynamics of sustainable transformations, making the sample size conducive to generating valid and insightful conclusions within this qualitative research framework.

Strategy: There's a need for clarity in terms of the main data collection strategy. D) Were the interviews structured, semi-structured, or unstructured? E) If secondary data was used, from what sources was this data collected, and how was it integrated with primary data? F) Why both methods applied, is it even neccessary, i dont think the two methods neccessarily fulfill main aim of triangulation.

PR - D) “Were the interviews structured, semi-structured, or unstructured”?

AU: Thank you very much for your comments. In the fourth paragraph of methodology, we clarify that the "A qualitative research methodology was employed, utilizing semi-structured techniques..."

PR - E) “If secondary data was used, from what sources was this data collected, and how was it integrated with primary data”?

AU: Thank you very much for your comments. Secondary data were obtained through Web of Summit data, Internet research, and literature reviews performed using Web of Science (WoS) and Scopus platforms. In sub-topic 3.1. is declared "Initially, we conducted a literature review on ATs, sustainable elements, and TBM in the context of start-ups using the Web of Science (WoS) and Scopus platforms..." and 3.2. "These start-ups were recognized either through data from the Web of Summit or through awards in Portugal and internationally, through internet research, covering the years 2014 to 2020."

The integration of secondary data with primary data occurred through the triangulation of findings, as explained in the methodology (see last paragraph). The secondary data from the literature review provided a theoretical framework and context for the study, while the primary data collected through interviews offered real-world insights and perspectives from start-ups. The qualitative analysis of the interviews and the Likert scale responses complemented and validated each other, reinforcing the qualitative findings with quantitative evidence. This integration allowed for a comprehensive understanding of the driving elements behind the sustainable transformation of business models within the start-up context.

PR - F) “Why both methods applied, is it even neccessary, i dont think the two methods neccessarily fulfill main aim of triangulation”.

AU: Thank you for sharing your considerations. Undoubtedly, within this research category, the integration of both survey and qualitative data collection methodologies serves distinct purposes, contributing to a more holistic comprehension of the research focus. As previously explained (and further detailed in the methodology section), the interviews played an essential role in corroborating the findings from secondary data analysis and in offering fresh insights (please refer to section 3.1's concluding paragraph). The "Mixed Methods Research" approach has been explored in diverse research studies.

Regarding the sample size, all sustainable start-ups in Portugal, representing the complete sample, were extended invitations to participate in the interview process. This approach yielded a participation rate of 26%, as elaborated in the concluding paragraph of sub-topic 3.4. As a result, this participation rate is considered satisfactory. Furthermore, in terms of the overall number of conducted interviews, the study suggests that future research endeavors could encompass broader geographic coverage to encompass a wider spectrum of start-ups. Your considerations are greatly appreciated.

  1. “The discussion is not connected to previous research in the field. The authors should link their findings with prior research, thereby placing their work in the broader context of the literature. Also consider adding some research that could enhance background such as Xing, Z., Huang, J., & Wang, J. (2023): Unleashing the potential: Exploring the nexus between low-carbon digital economy and regional economic-social development in China." And Zhang, Z., Hao, L., Linghu, Y., & Yi, H. (2023): "Research on the energy poverty reduction effects of green finance in the context of economic policy uncertainty."

"Clustered institutional investors, shared ESG preferences and low-carbon innovation in family firm;, Zhao, S., Zhang, L., An, H., Peng, L., Zhou, H.,... Hu, F. (2023): "Has China's low-carbon strategy pushed forward the digital transformation of manufacturing enterprises? Evidence from the low-carbon city pilot policy."

AU: We extend our gratitude for your careful review and insightful comments. We appreciate your perspective, and it's important to note that our study does indeed have connections to previous research. To illustrate this, let's delve into specific sub-topics: In sub-topic 2.1 of our "Conceptual Framework," we explore the STT. This theory sheds light on how it can empower transformative start-up business models to contribute to decarbonization efforts.

Moving to sub-topic 2.2, we delve into SNM. This section elaborates on how start-ups can play a catalytic role by introducing innovative changes within specific niches, fostering transformation.

Transitioning to sub-topic 2.3, we engage extensively with the concept of TBM for decarbonization. This segment thoroughly explores the significance of TBM in driving sustainable shifts.

In sub-topic 2.4, our focus shifts to the role of ATs in transformative start-up BMs. Notably, we highlight the pivotal role of AI-ML in relation to sustainability and the transformation of business models.

These are examples of sub-topics that collectively reinforce the connections between our study and previous research, offering a comprehensive understanding of how theoretical frameworks align with practical insights in the domain of sustainable and TBMs.

Your suggestion to link our findings with prior research and contextualize our work within the existing literature is highly valued. We appreciate the reference suggestions you provided, including the works by Xing, Z., Huang, J., & Wang, J. (2023) and Zhang, Z., Hao, L., Linghu, Y., & Yi, H. (2023), as well as Zhao, S., Zhang, L., An, H., Peng, L., Zhou, H., Hu, F. (2023). These studies indeed offer valuable perspectives in the realm of low-carbon innovation and digital transformation. However, it is our intention to maintain a focused examination of TBMs that foster sustainable practices, specifically within the start-up context. Our aim is to delve into the evolution of these TBMs as they pertain to start-ups. While we acknowledge the relevance of the references you provided, we believe that our study's specific scope and objectives warrant an emphasis on TBMs within the start-up domain. Your feedback has greatly contributed to the refinement of our work, and we are appreciative of your thoughtful considerations.

  1. “The scientific rigor of the study appears uncertain. The paper combines various literature streams, like Sustainability Transitions Theory (STT), Strategic Niche Management (SNM), and the Business Models approach. However, the integration of these streams should be carried out more systematically. The application of these theoretical frameworks should be more apparent in the analysis and findings”.

AU: Thank you very much for your comments. We appreciate your insights, and we certainly concur with your observations. In response, we have revised to enhance the coherence of topic 4.1, ensuring a stronger connection to STT. Additionally, we have introduced a new paragraph within sub-topic 4.2.1 to establish a meaningful link to TBM (as seen in the second paragraph), while also elaborating on the significance of the SNM approach (as evident in the concluding paragraph). Your feedback has been instrumental in refining the presentation of our work.

  1. “Are theoretical contributions evident”?

AU: Thank you very much for your comments. Yes, the theoretical contributions are evident in the article, particularly in the results section. The text discusses several theoretical contributions that have emerged from the analysis and findings of the study. For example, the article discusses theoretical contributions related to the integration of some elements of technology and sustainability in business models, the role of new institutional arrangements, the potential of specific business model shifts, and the practical implications for both organizations and policymakers. These contributions demonstrate how the study extends existing theoretical frameworks and provides insights into the transformation of business models for sustainability.

  1. “Most of the tables and figures report on issues that do not require individual visual representation, so the tables should be merged, and figures”.

AU: We sincerely appreciate your comments. Your perspective regarding the utilization of tables and figures is noted, and we value your input. It is our belief, however, that the tables and figures presented in their current format serve to effectively illustrate and highlight the crucial aspects of the research. While we acknowledge the potential for merging tables and figures, we have chosen to maintain the current presentation style as we consider it to be appropriate and beneficial for conveying the intricate details and nuances inherent to the subject matter. Your considerations have contributed to a thoughtful evaluation of our content, and we wish to express our gratitude for your valuable feedback.

Reviewer 2 Report

The recommendations are as follows

1. What is decarbonisation? Please give a description in the text. Also, what are the factors that affect decarbonisation? Please give it by the author as well. Further, what is the scope of research on decarbonisation? Please also give it.

2, Please give the feasibility of the grounded Theory and Likert scale used in the study and the logical relationship between the factors studied and decarbonisation.

3, The analysis of the paper needs to be deeper and please give the future countermeasures and suggestions to improve the effectiveness of decarbonisation of enterprises.

No comment

Author Response

Dear reviewer

We sincerely thank you very much for your thoughtful review. Your feedback has been invaluable to us, and we greatly appreciate the insights you've provided. Incorporating your constructive comments, we have revised the article. For your convenience, all revisions have been highlighted in blue. We believe these modifications address the specific areas you highlighted and enhance the overall quality of the manuscript. We are optimistic that these revisions will contribute to the successful publication of our article. Below, you will find your recommendations along with our corresponding responses:

  1. “What is decarbonisation? Please give a description in the text. Also, what are the factors that affect decarbonisation? Please give it by the author as well. Further, what is the scope of research on decarbonisation? Please also give it”.

Authors (AU): Thank you very much for the comments. We added a part in the conceptual framework to clarify this point.

  1. “Please give the feasibility of the grounded Theory and Likert scaleused in the study and the logical relationship between the factors studied and decarbonisation”.

AU: Thank you very much for your comments. Certainly, the utilization of Grounded Theory and Likert scale methodologies in our study holds notable feasibility. Grounded Theory, as a systematic and inductive approach, allows us to develop hypotheses and theories rooted in empirical evidence, thereby facilitating a comprehensive exploration of the intricate dynamics within start-up business models. Likewise, the Likert scale method provides a quantifiable framework for participants to express their perspectives on the challenges and importance of several factors. This combination enables us to gain both in-depth qualitative insights and quantitative data, fostering a holistic understanding of our research domain.

Regarding the logical relationship between the factors under study and decarbonization, our work delves into transformative business models (TBMs) and their role in sustainable practices. While our primary objective is to identify key elements characterizing TBMs for sustainability, we acknowledge that these models inherently have the potential to drive decarbonization efforts. Sustainable practices, integral to TBMs, encompass resource efficiency, technological innovation, and environmental responsibility. As such, TBMs can effectively contribute to decarbonization by reducing emissions through innovative processes, optimizing resource use, and promoting environmentally conscious business strategies. This linkage between TBMs and decarbonization underscores the critical role of TBMs in advancing the transition towards a low-carbon economy.

  1. “The analysis of the paper needs to be deeper and please give the future countermeasures and suggestions to improve the effectiveness of decarbonisation of enterprises”.

AU: We would like to express our gratitude for your constructive feedback, which has led us to implement several revisions that have undoubtedly enhanced the clarity and consistency of our article. We appreciate your suggestion to delve deeper into the analysis and offer future countermeasures and suggestions to enhance the efficacy of decarbonization strategies in businesses. We have meticulously considered your viewpoint, and while we recognize the potential advantages of expanding on these aspects, we remain confident that our current approach effectively addresses our research objectives and contributes significantly to the field. We have taken great care to ensure that the article maintains its coherence and relevance within the defined scope of our study. Your thoughtful comments and involvement with our work are appreciated.

Reviewer 3 Report

Authors focus on studying the business model of start-ups in the situation of technology and decarbonization development. First, I would ask authors for the definition of start-ups, and as well as for providing an explanation of strategy of start-ups. Do you think that each enterprise having less than 10 employees and existing three years on market is a start-up? What are unique characteristics of start-ups? What makes them different from a business as usual?

I would agree that Airbnb, Uber, Netflix, Spotify, and Zipcar created new business model, but they are not start-ups, I suppose, could you compare start-ups with them? Maybe yes, maybe not, I do not exclude that there are some important characteristics, which are similar, for example usage of technology.

Through the interviewing, Authors characterized companies, their activities and attitudes towards technology. For sure, the study is interesting and important. However, I am afraid authors have not sufficiently well emphasized, presented, and described the new business models. What novelty do the studied companies propose?

Literature review is appropriate. I would like for clear specification of research questions and hypotheses. They are not clear.  The methodology also is not clear.

Author Response

Dear reviewer

We sincerely thank you very much for your thoughtful review. Your feedback has been invaluable to us, and we greatly appreciate the insights you've provided. Incorporating your constructive comments, we have revised the article. For your convenience, all revisions have been highlighted in blue. We believe these modifications address the specific areas you highlighted and enhance the overall quality of the manuscript. We are optimistic that these revisions will contribute to the successful publication of our article. Below, you will find your recommendations along with our corresponding responses:

PR: “Authors focus on studying the business model of start-ups in the situation of technology and decarbonization development. First, I would ask authors for the definition of start-ups, and as well as for providing an explanation of strategy of start-ups. Do you think that each enterprise having less than 10 employees and existing three years on market is a start-up? What are unique characteristics of start-ups? What makes them different from a business as usual”?

Authors (AU): Thank you very much for the comments. The study seeks to comprehend how organizations can facilitate the shift towards sustainable practices and decarbonization. In pursuit of this comprehension, it examines transformative business models within start-ups from the years 2014 to 2020. In fact, there are several definitions for the term start-up. Consistent with the approach of other authors, we included within the methodology the definition of the term "start-up" for the current study and highlighted distinctions between start-ups and other enterprises, such as traditional companies. Your valuable feedback prompted us to improve the concept of a start-up (emphasis in blue color).

PR: “I would agree that Airbnb, Uber, Netflix, Spotify, and Zipcar created new business model, but they are not start-ups, I suppose, could you compare start-ups with them? Maybe yes, maybe not, I do not exclude that there are some important characteristics, which are similar, for example usage of technology”.

AU: Thank you very much for your comments. We added a part (blue color) to make the subject clearer.

PR: “Through the interviewing, Authors characterized companies, their activities and attitudes towards technology. For sure, the study is interesting and important. However, I am afraid authors have not sufficiently well emphasized, presented, and described the new business models. What novelty do the studied companies propose”?

AU: Thank you very much for your comments. We have added (Blue color) a paragraph in the section of results which has contributed to making the study clearer.

PR: “Literature review is appropriate. I would like for clear specification of research questions and hypotheses. They are not clear. The methodology also is not clear”.

AU: Thank you very much for your comments. We made changes to the introduction, methodology, and conclusion of the article to clarify the subject and improve the study.

Round 2

Reviewer 1 Report

While improvements have been made and the comments addressed with satisfactory explanations, the paper still requires major revisions to enhance its overall quality. Please center your paper around a single methodology, consolidate graphs where possible, and ensure all changes are highlighted in red. The structure and coherence of the paper need further refinement.

You have addressed some limitations in the comments, but please include a dedicated section for 'Limitations and Future Studies' in the paper.

proofread

Author Response

Dear reviewer #1

We express our heartfelt gratitude for your valuable feedback during the second round of review. Taking into consideration your constructive comments, we have diligently revised the article once again. To make it easier for you to identify the changes, we have highlighted all revisions in red. We firmly believe that these modifications effectively address the specific areas you pointed out, ultimately enhancing the overall quality of the manuscript. We are confident that these revisions will significantly enhance our article's chances of successful publication. Please find below a summary of their recommendations, along with our corresponding responses:

1. We group six figures into a single group, which can be referenced as Figure 4.

2. Regarding the limitations and future studies, as recommended, we have incorporated a dedicated section, highlighted in red within the article, to address the 'Limitations and Future Studies' pertaining to our research.

3. The methodology employed in this study required mixed methods and techniques to achieve the research objectives, as it integrates the collection and analysis of both quantitative and qualitative data. It is essential to highlight that we adopted Grounded Theory, which is a systematic and inductive qualitative research method, to develop hypotheses and theories rooted in empirical evidence. This method enabled us to delve deeper into the phenomena under investigation, offering valuable insights into social relationships and group behavior.

To enhance the robustness and consistency of our analysis, we have complemented Grounded Theory with a Likert scale. This amalgamation of methods not only validated our qualitative findings, but also quantified responses pertaining to the key themes and categories identified during our qualitative analysis. This mixed-methods approach affords us a comprehensive understanding of the driving forces behind TSBMs.

It is noteworthy that, as evidenced in our first-round feedback, several other authors (as exemplified below) have also embraced a similar methodology, underscoring the validity and effectiveness of this approach in research:

  • https://doi.org/10.1016/j.tra.2022.05.011
  • https://doi.org/10.1016/j.tra.2016.11.012
  • https://doi.org/10.1057/ejis.2014.23
  • https://doi.org/10.3390/su11030844
  • https://doi.org/10.1016/j.rser.2022.112449
  • https://catalyst.harvard.edu/community-engagement/mmr/

4. Regarding the structure and coherence of the article, after carefully reviewing your feedback and considering the structure of our paper, we believe that our manuscript is well-structured and maintains a high level of coherence. Here are some key points to support this assessment:

4.1. Logical Flow: Our paper follows a logical progression from the introduction through the conceptual framework, methodology, results, and conclusion. This flow allows readers to follow the development of our research from the initial introduction of the topic to the presentation of findings and their implications.

4.2. Clear Subsection Organization: We have organized our content into clearly delineated subsections, making it easy for readers to navigate the paper and locate specific information. Each subsection serves a specific purpose and contributes to the overall understanding of our research.

4.3. Detailed Conceptual Framework: The "Conceptual Framework" section provides a comprehensive foundation for our study, breaking down the theoretical underpinnings into manageable subsections. This approach helps readers grasp the theoretical context without becoming overwhelmed by excessive detail.

4.4. Methodological Transparency: Our methodology section is detailed and transparent, ensuring that readers have a clear understanding of our research methods and data collection procedures. This transparency is essential for the credibility of our research.

4.5. Results Presentation: In the "Results" section, we have structured our findings logically, with separate subsections for secondary data analysis and interview results. This organization allows us to present our results in a clear and coherent manner.

4.6. Conclusion and Further Steps: The "Conclusions" section includes subsections for discussing the limitations of our study and providing recommendations for future research (as recommended in separated topics). These components are vital for acknowledging the scope of our work and guiding future research directions.

While we acknowledge the detailed nature of our methodology section, we believe that it is essential to ensure transparency and rigor in our research. However, we will review the subsections you mentioned to ensure that they do not unnecessarily duplicate information and are presented concisely.

Therefore, we are confident that the structure and coherence of our article are well-established, aligning with the standards of academic publishing. We appreciate your valuable feedback and will make the necessary revisions to address any specific concerns you have raised while preserving the overall structure and coherence of the manuscript.

5. We conducted a thorough revision of the English language, and the adjusted parts have been highlighted in red for clarity and improved readability.

Thank you very much. Your thoughtful insights are highly appreciated.

Round 3

Reviewer 1 Report

can further work on paper

no comments